# tRNA sequences can assemble into a replicator

**Alexandra Kühnlein†, Simon A Lanzmich†, Dieter Braun\***

Systems Biophysics, Physics Department, Center for NanoScience, Ludwig-Maximilians-Universität München, Munich, Germany

**Abstract** Can replication and translation emerge in a single mechanism via self-assembly? The key molecule, transfer RNA (tRNA), is one of the most ancient molecules and contains the genetic code. Our experiments show how a pool of oligonucleotides, adapted with minor mutations from tRNA, spontaneously formed molecular assemblies and replicated information autonomously using only reversible hybridization under thermal oscillations. The pool of cross-complementary hairpins self-selected by agglomeration and sedimentation. The metastable DNA hairpins bound to a template and then interconnected by hybridization. Thermal oscillations separated replicates from their templates and drove an exponential, cross-catalytic replication. The molecular assembly could encode and replicate binary sequences with a replication fidelity corresponding to 85–90 % per nucleotide. The replication by a self-assembly of tRNA-like sequences suggests that early forms of tRNA could have been involved in molecular replication. This would link the evolution of translation to a mechanism of molecular replication.

## Introduction

A machine to create replicate of itself is an old dream of engineering (*von Neumann, 1951*). Biological systems have solved this problem long ago at the nanoscale with DNA and RNA. Their replication machinery was optimized to perfection through Darwinian evolution. In modern living systems, the replication of DNA and RNA necessitates the formation of covalent bonds. It requires an interconnected machinery: proteins need to perform base-by-base replication of sequence information, a modern metabolism to supply activated molecules, and tRNA as well as the ribosome to create the required proteins.

This is a complex system to set up in the first place at the emergence of life. The RNA world hypothesis proposes, that early on, the catalytic function of highly defined RNA sequences was used for self-replication (*Horning and Joyce, 2016*; *Orgel, 2004*; *Turk et al., 2011*). These ribozymes catalyze the ligation of RNA (*Doudna et al., 1991*; *Mutschler et al., 2015*; *Paul and Joyce, 2002*; *Robertson et al., 2001*; *Walton et al., 2020*) and the addition of individual bases (*Attwater et al., 2013*; *Horning and Joyce, 2016*). These very special sequences were engineered using in vitro evolution. It is unclear how autonomous evolution of early life could have reached such levels of sequence complexity.

Here, we focus on how such replication may have been predated by simpler forms of self-replication. Creating a replicator must fulfill a series of requirements. Replication must yield fidelity in copying, be fast, enable exponential replication, be fed by an autonomous energy source, not require complex sequences and should not form too many replicates without the existence of a template.

We show that replication of information can be realized by the reversible hybridization interactions between tRNA-like molecules alone. The proposed mechanism is driven by an external physical non-equilibrium setting, in our case thermal oscillations. Since the process does not involve chemical ligation, it does not rely on a particular non-enzymatic or catalytic ligation chemistry (*Dolinnaya et al., 1988*; *Engelhart et al., 2012*; *Patzke et al., 2014*; *Pino et al., 2011*;

**\*For correspondence:**
dieter.braun@lmu.de

†These authors contributed equally to this work

**Competing interests:** The authors declare that no competing interests exist.

**eLife digest** The genetic code stored within DNA contains the instructions for manufacturing all the proteins organisms need to develop, grow and survive. This requires molecular machines that 'transcribe' regions of the genetic code into RNA molecules which are then 'translated' into the string of amino acids that form the final protein. However, these molecular machines and other proteins are also needed to replicate and synthesize the sequences stored in DNA. This presents evolutionary biologists with a 'chicken-and-egg' situation: which came first, the DNA sequences needed to manufacture proteins or the proteins needed to transcribe and translate DNA?

Understanding the order in which DNA replication and protein translation evolved is challenging as these processes are tightly intertwined in modern-day species. One theory, known as the 'RNA world hypothesis', suggests that all life on Earth began with a single RNA molecule that was able to make copies of itself, as DNA does today. To investigate this hypothesis, Kühnlein, Lanzmich and Braun studied a molecule called transfer RNA (or tRNA for short) which is responsible for translating RNA into proteins. tRNA is assumed to be one of the earliest evolved molecules in biology. Yet, why it was present in early life forms before it was needed for translation still remained somewhat of a mystery.

To gain a better understanding of tRNA's role early in evolution, Kühnlein, Lanzmich and Braun made small changes to its genetic code and then carried out tests on these tRNA-like sequences. The experiments showed these 'early' forms of tRNA can actually self-assemble into a molecule which is capable of replicating the information stored in its sequence. It suggests early forms of tRNA could have been involved in replication before modern tRNA developed its role in protein translation.

With these experiments, Kühnlein, Lanzmich and Braun have identified a possible evolutionary link between DNA replication and protein translation, suggesting the two processes emerged through one shared pathway: tRNA. This deepens our understanding about the origins of early life, while taking biochemists one step closer to their distant goal of recreating self-replicating molecular machines in the laboratory.

*Rohatgi et al., 1996*; *Sievers and von Kiedrowski, 1994*; *von Kiedrowski, 1986*) or particular catalytically active sequences, but merely requires sequence complementarity. The advantage of reversible hybridization is the re-usability of educts and products. Moreover, sequence-encoded interactions can self-select by forming agglomerates.

Nature's approach to achieve exponential growth is the usage of cross-catalysis: the replicate of a template serves as a template for the next round of replication. For short replicators under isothermal conditions, the binding between template and replicate has to be weak such that the dissociation of strands happens spontaneously and is not rate limiting (*Paul and Joyce, 2002*; *Sievers and von Kiedrowski, 1994*; *von Kiedrowski, 1986*). For longer replicates, temperature change has successfully been used to separate strands for replication catalyzed by thermostable proteins (*Barany, 1991*; *Saiki et al., 1985*). For catalytic RNA, elevated salt concentrations disfavor strand separation by temperature and catalyze hydrolysis (*Horning and Joyce, 2016*). In an interesting alternative to strand separation by temperature, Schulman et al. used moderate shear flows to separate DNA tile assemblies (*Schulman et al., 2012*).

Apart from nucleotide-based replicators, very interesting replication systems using non-covalent interactions have been developed with non-biological compounds (*Bottero et al., 2016*; *Sadownik and Philp, 2008*; *Tjivikua et al., 1990*), peptide-based approaches (*Altay et al., 2017*; *Bourbo et al., 2011*; *Carnall et al., 2010*; *Lee et al., 1996*; *Rubinov et al., 2012*), and peptide nucleic acids (*Ura et al., 2009*). We also want to point to several instructive reviews about the state-of-the-art systems chemistry regarding self-replication (*Adamski et al., 2020*; *Ashkenasy et al., 2017*; *Kosikova and Philp, 2017*).

In the past, metastable hairpin states have been prepared in a physically separated manner. The reaction was then triggered by mixing. For example, the mixing of hairpins with a trigger sequence has been shown to form long concatemers (*Dirks and Pierce, 2004*). With a similar logic, mixing a low entropy combination of molecules was used to create entropically driven DNA machines,

including exponentially amplifying assemblies (*Zhang et al., 2007*). These reactions run downwards into the binding equilibrium. However, the preparation of the initial low entropy state required human intervention or a unique flow setting for mixing.

## Sequence design

We designed a set of cooperatively replicating DNA strands using the program package NUPACK (*Zadeh et al., 2011*). The sequences are designed to have self-complementary double hairpins and are pairwise complementary within the molecule pool, such that the 3′ hairpin of one strand is complementary to the 5′ hairpin of the next. Their structure resembles the secondary structure of proto-tRNAs proposed by stereochemical theories (*Figure 1a*), comprising two hairpin loops that surround the anticodon with a few neighboring bases (*Krammer et al., 2012*). The lengths of 82–84 nt of the double hairpins are that of average tRNA molecules (*Sharp et al., 1985*), with stem loops consisting of 30–33 nt and the information-encoding interjacent domains of 15 nt. As the replication mechanism is based on hybridization only, it is expected to perform equally well for DNA and RNA. Here, we implemented the system with DNA and not RNA as done previously (*Krammer et al., 2012*). Both, in the design and the implementation we did not see significant differences between the two versions. Because of the simpler and more inexpensive synthesis of the 82–84 nt long sequences we now implemented the replicator in DNA. Due to short heating times and moderate magnesium concentrations, we estimate that an RNA version could survive for days if not weeks (*Li and Breaker, 1999*; *Mariani et al., 2018*). The most critical step regarding the RNA stability would be the initial temperature spike to 95 °C, which remains unchanged from our previous study (*Krammer et al., 2012*) and did not prove critical. We also show that an RNA version behaves structurally identical to the implemented DNA version (*Figure 1—figure supplement 1*).

## Replication mechanism

The replication mechanism is a template-based replication, where instead of single nucleotides, information is encoded by a succession of oligomers. The domain, at the location of the anticodon in tRNA, is the template sequence and thus contains the information to be replicated. We therefore term it information domain. The goal is to replicate the succession of information domains.

To allow longer replicates, we chose the resulting meta-sequences to be periodic with a periodicity of four different hairpins. This makes the minimal cyclic meta-sequence large enough to keep the information domains accessible even in cyclic configuration. The information domains feature a binary system and contain sequences marked by '0' and '1' (blue/red). For replication, two sets of strands replicate strings of codons in a cross-catalytic manner (*Figure 1b*), using complementary information domains (light/dark colors).

The replication is driven by thermal oscillations and operates in four steps (*Figure 1b*): (0) Fast cooling within seconds brings the strands to their activated state with both hairpins closed. (1) At the base temperature, activated strands with complementary information domains can bind to an already assembled template. (2) Thermal fluctuations cause open-close fluctuations of the hairpins. When strands are already bound to a template at the information domain, those fluctuations permit adjacent complementary hairpins of different strands to bind. In this way, the succession of information domains is replicated. (3) Subsequent heating splits the newly formed replicate from the template at the information domains. Due to their higher melting temperatures, the backbone of hairpin strands remains stable. Both, replicate and template, are available for a new replication round. This makes both the replicate and the template replication cross-catalytic in a subsequent step. Later, high temperatures spikes can unbind and recycle all molecules for new rounds of replication.

Because of the initial fast cooling, all hairpins are closed in free solution. This inhibits the formation of replicates without template. While the binding of adjacent hairpins with template happens within minutes, hairpins in free solution connect without template only on timescales slower than hours and thus give false positives at a very low rate.

The basic principle of this replication mechanism was previously explored by Krammer et al. using a set of four hairpins using half a tRNA sequence (36 nt) that amplified into dimers (*Krammer et al., 2012*). This amplification could not encode information and suffered from a high rate (>50 %) of unspecific amplification without template (Figure 4 therein). Here, in contrast, we demonstrate exponential amplification, and the replicator can now encode sequence information '0' and '1' with four

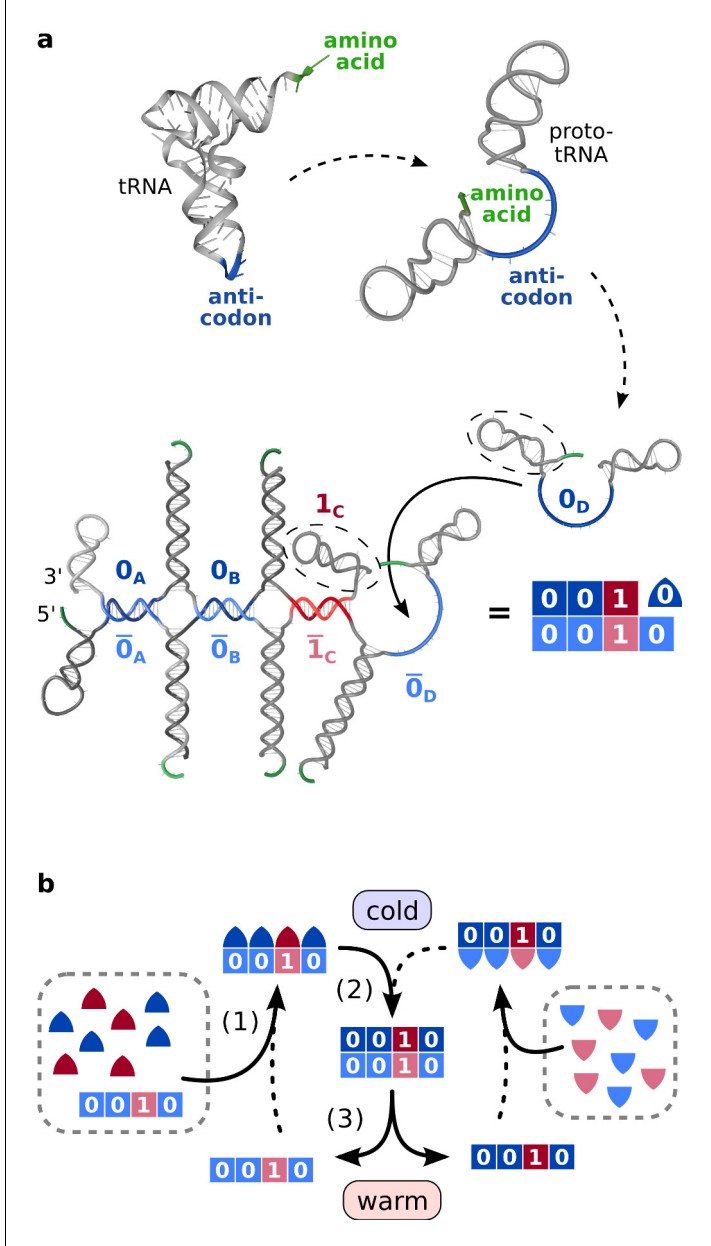

**Figure 1.** Heat-driven replication by hybridization using hairpin structures inspired from transfer RNA. (**a**) Transfer RNA folds into a double-hairpin conformation upon very few base substitutions. In that configuration, the 3'-terminal amino acid binding site (green) is close to the anticodon (blue) and a double hairpin structure forms. A set of pairwise complementary double hairpins can encode and replicate sequences of information. A binary code implemented in the position of the anti-codon, the information domain, allows to encode and replicate binary sequences (red vs blue). Each strand (82-84 nt) comprises two hairpin loops (gray) and an interjacent unpaired information domain of 15 nt length (blue/red, here: $0_D$). The displayed structure of eight strands shows replication of a template corresponding to the binary code 0010. Note, that no covalent linkage is involved in the process. (**b**) Replication is driven by thermal oscillations in four steps: (0) The hairpins are activated into their closed conformation by fast cooling indicated by triangles. (1) Strands with matching information domain bind to the template. (2) Fluctuations in the bound strands' hairpins facilitate the hybridization of neighboring strands. (3) Subsequent heating splits replica from template, while keeping the longer hairpin sequences connected, freeing both as templates for the next cycle.

The online version of this article includes the following figure supplement(s) for figure 1:

**Figure supplement 1.** Secondary structure predictions and free energy calculations for the replicator in DNA and RNA using NUPACK.

bits. Moreover, the strands making up the new replicator are double hairpins with the sequence structure and length of tRNA. The replicator now shows a significantly decreased unspecific amplification without template of approximately 10 % (Figure 5a).

## Results

### Analysis of molecule conformations

Native polyacrylamide gel electrophoresis (PAGE) showed that the double hairpins assembled as intended (*Figure 2*). Comparing different subsets of strands allowed to identify all gel bands.

All complexes were formed at concentrations of 200 nM of each strand and could be resolved despite their branched tertiary structure. Friction coefficients of complexes of two to four strands were 1.6–1.8-fold higher than for linear dsDNA, and 2.4-fold higher for larger complexes (4:4 configuration, ca. 660 nt, *Figure 2—figure supplement 1*). This agrees with the branched structure of the suggested strand assembly geometry (*Figure 1a*). Partially assembled complexes of two or three strands bound to a four-strand template could be resolved (*Figure 6—figure supplement 1*). Complexes containing single bound information domains were not stable during electrophoresis (*Figure 2*, lanes 2, 7 and *Figure 6—figure supplement 1*). This allowed to differentiate fully assembled complexes from those where individual strands are bound to a template but have not formed

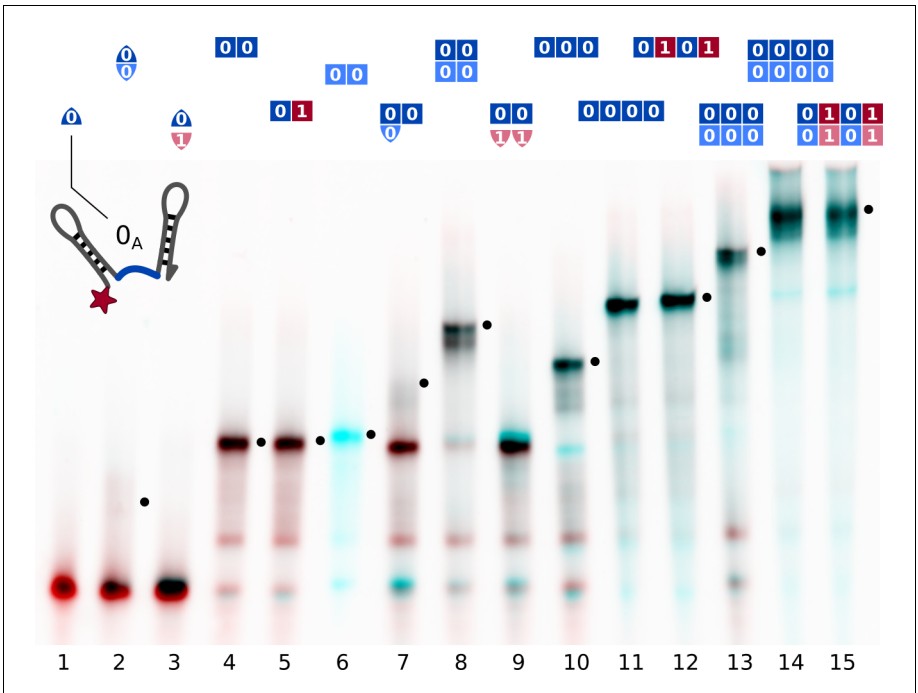

**Figure 2.** Assembly of different subsets of the cross-replicating system of strands observed by native gel electrophoresis. Samples contained strands at 200 nM concentration each and were slowly annealed as described in Materials and methods. Lane contents are indicated at the top of each lane. Comparison of different lanes allowed for the attribution of bands to complexes. Complexes incorporating all present strands are marked (•). The red channel shows the intensity $0_A$-Cy5, the cyan channel shows SYBR Green I fluorescence. Single information domain bonds (lane 2, 7) break during gel electrophoresis.

The online version of this article includes the following source data and figure supplement(s) for figure 2:

**Source data 1.** Source data for assembly of different subsets of the cross-replicating system of strands observed by native gel electrophoresis.

**Figure supplement 1.** Gel mobilities of different complexes compared to linear dsDNA.

**Figure supplement 1—source data 1.** Source data for gel mobilities of different complexes compared to linear dsDNA.

backbone duplexes. Covalent end labels and two reference lanes on each gel were used to quantify concentrations from gel intensities using image analysis as described in Materials and methods.

## Selection by agglomeration and sedimentation

For a replicator to be autonomous, there must be a mechanism in place to select, assemble and (re-) accumulate its molecular components purely at one location. We argue that DNA hydrogels could offer such a solution. While DNA often, also in our case, assembles into agglomerates, DNA hydro-gels have been shown to be able to form fluid phases if gaps of single bases were added to create flexible linkers between molecules (*Nguyen and Saleh, 2017*).

We combined eight matching hairpin sequences of design as introduced in *Figure 1* at moderately elevated concentrations and cooled the system to only 25 °C after separating the molecules at 95 °C (*Figure 3*). We found the spontaneous formation of agglomerates that were large enough to sediment under gravity. The initial homogeneous fluorescence turned into micrometer-sized grains and sedimented within hours. The fluorescence was provided by a covalently attached label to either strand $0_A$ or $1_A$. Since the double hairpins have a periodic boundary condition, they can create large assemblies (*Figure 3a*).

It is evident from *Figure 3—video 1* that the sedimentation was very selective. When only seven of the eight matching hairpins were present, sedimentation was much weaker and, in most cases, undetectable (*Figure 3b,c*). For the full system, the sedimentation kinetics showed to be strongly concentration dependent (*Figure 3—figure supplement 1b*). Analogous experiments with random sequences (random pool of 84 nt strands) at equal concentration did not show agglomeration nor sedimentation (*Figure 3—figure supplement 1c*). We have previously found that similar hairpin molecules provided the shortest sequences capable of forming agglomerates (*Morasch et al., 2016*).

The above results suggest that agglomeration could serve as an efficient way to assemble matching hairpins from much less structured and selected sequences in an autonomous way. After the molecules have been assembled as sedimented agglomerates, a convection flow can carry the large assemblies into regions of warmer temperatures, where the molecules would be disassembled by heat and activated for replication with a cooling step. Similar recycling behavior is seen in thermal gradient traps (*Morasch et al., 2016*), which were also found to enhance the molecular assembly (*Mast et al., 2013*) with characteristics that can match the above scenario.

## Templating kinetics

Hybridization between stems of neighboring hairpins (*Figure 1b*, step 2) was catalyzed by the presence of already assembled complexes $\bar{0}_A\bar{0}_B\bar{0}_C\bar{0}_D$, confirming its role as a template. Assembly kinetics at 45 °C were recorded in reactions containing 200 nM of each strand for a range of template concentrations. At 120 nM template concentration, 40 % yield was achieved within 10 min (*Figure 4b*, black line). The untemplated, spontaneous reaction proceeded significantly slower (1.4 % yield, light gray line).

Assembly rates showed a strong dependence on incubation temperature (*Figure 4c*). At 39 °C, the reaction proceeded significantly slower than at 42 °C or 45 °C. This is because the hairpins are predominantly in closed configuration and cannot bind to neighboring molecules in the assembly. Binding between complementary information domains still occurs, but the formation of bonds between neighboring strands becomes rate limiting. Above the melting temperature of the information domain (48 °C) (see *Figure 4—figure supplement 1*), template-directed assembly becomes slower. However, the slower kinetics of template-directed product formation are partially superposed by the spontaneous product formation lacking an initial template (*Figure 4c*, small circles), which becomes an additional reaction channel due to the now open hairpins.

## Exponential amplification

As intermediate step toward replication, we studied amplification reactions under thermal oscillations (*Figure 5*). The amplification reactions only contained strands encoding for information domain '0', that is $0_A$, $\bar{0}_A$, $0_B$, $\bar{0}_B$, ..., $\bar{0}_D$. The strands were subjected to thermal oscillations between $T_{base}$ = 45 °C and $T_{peak}$ = 67 °C. The lower temperature was held for 20 min, the upper for one second with temperature ramps amounting to 20±1 s in each full cycle. This asymmetric shape of the temperature cycle accords with differences in kinetics of the elongation step and the melting of the information

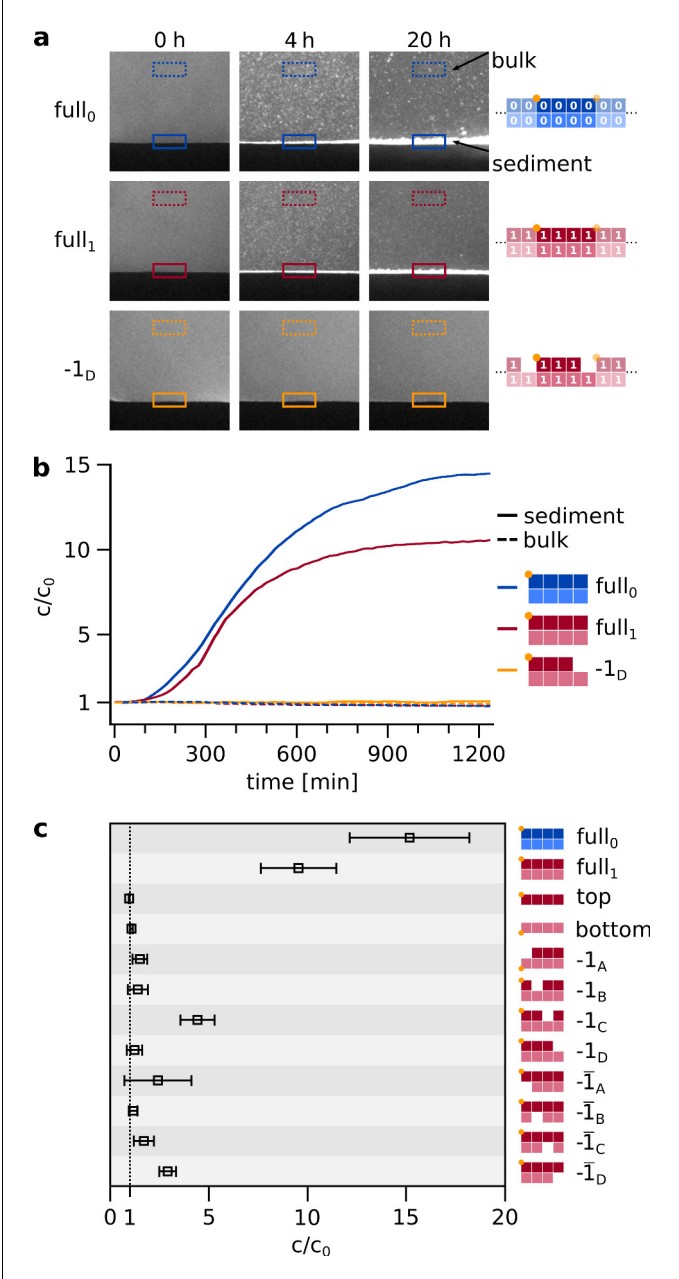

**Figure 3.** Spontaneous self-assembly and sedimentation of matching hairpins. (a) In a simple, sealed microfluidic chamber (*Figure 3—figure supplement 2*), the hairpin strands can self-assemble into agglomerates and sediment on a timescale of hours. The sample was initially heated to 95 °C for 10 s to ensure an unbound initial state, then rapidly (within 30 s) cooled to 25 °C, where self-assembly and sedimentation occured. Note, that agglomeration and sedimentation only occured if all eight matching hairpins were provided (top two rows) but not in the case of a knockout ($-1_D$, bottom row). For quantification, the bulk and sediment intensities were normalized by the first frame after heating. Samples contained strands at total concentration of 5 μM, about threefold higher than in *Figure 2* and the following replication experiments. (b) Time traces of concentration increase for sediment and bulk of different configurations, same examples as shown in a. The time traces of all further knockout permutations are shown in *Figure 3—figure supplement 1b*. (c) Final concentration increase of sediment, relative to first frame after heating, for all configurations. The final values (N≥3) for $c/c_0$ are retrieved from fitting the time traces. For the full set of complementary hairpins, self-assembly and sedimentation is most pronounced.

The online version of this article includes the following video, source data, and figure supplement(s) for figure 3:

**Source data 1.** Source data for spontaneous self-assembly and sedimentation of matching hairpins.

**Figure supplement 1.** Extended data on self-assembly and sedimentation.

*Figure 3 continued on next page*

*Figure 3 continued*

**Figure supplement 1—source data 1.** Source data for extended data on self-assembly and sedimentation.
**Figure supplement 2.** Sketch of microfluidic chamber.
**Figure 3—video 1.** Sedimentation of DNA agglomerates.
https://elifesciences.org/articles/63431#fig3video1

domain. It is typical for trajectories in thermal convection settings with local heating (***Braun et al., 2003***).

The growth of molecular assemblies with different initial concentrations of template $\bar{0}_A\bar{0}_B\bar{0}_C\bar{0}_D$ revealed an almost linear dependence of the reaction velocity on the initial amount of template (***Figure 5a, b***). This confirms the exponential nature of the replication. The cross-catalytic replication kinetics can be described by a simplistic model that only considers the concentrations $c(t)$ of the template $0_A0_B0_C0_D$ and its complement $\bar{c}(t)$ of $\bar{0}_A\bar{0}_B\bar{0}_C\bar{0}_D$:

$$\frac{d}{dt}c(t) = k \cdot \bar{c}(t) + k_0 \, , \, \frac{d}{dt}\bar{c}(t) = k \cdot c(t) + k_0 \quad (1)$$

Here, $k$ is the rate of cross-catalysis and $k_0$ the spontaneous formation rate. For $c(t) \approx \bar{c}(t)$, the model corresponds to simple exponential growth on a per-cycle basis. The model can be solved in closed form but does not account for saturation effects from the depletion of monomers. Therefore, it is not valid for concentrations similar to the total concentration of each strand. Fitting the model to the amplification reactions with 0–45 nM of template $\bar{0}_A\bar{0}_B\bar{0}_C\bar{0}_D$ revealed rate constants of $k = 0.16$ cycle$^{-1}$ and $k_0 = 0.4$ nM cycle$^{-1}$ (***Figure 5b***). Amplification was robust with regard to the peak temperature of the oscillations. For $T_{peak}$ below 74 °C, the reaction remained almost unaffected (***Figure 5c***). Above, the temperature is too close to the melting transitions of the hairpin-hairpin duplexes, ranging from 76 to 79 °C (***Figure 4—figure supplement 1***).

The ability to withstand consecutive dilutions is characteristic for exponentially growing replicators and was tested for in serial transfer experiments. Strands encoding for '0' (i.e. $0_A$, $\bar{0}_A$, $0_B$, etc.) were thermally cycled with 30 nM of template $\bar{0}_A\bar{0}_B\bar{0}_C\bar{0}_D$. After three cycles each, samples were diluted one to one with buffer containing all eight strands as monomers at 200 nM each (***Figure 5d***). This high frequency of dilutions prevented the reaction from transitioning into the saturating regime. The cross-catalytic model was fitted to the data with the dilution factor as single free parameter, that was found

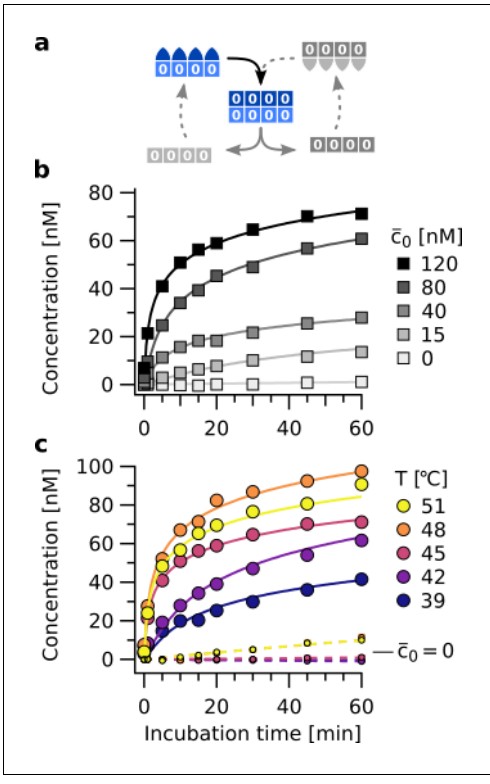

**Figure 4.** Isothermal template assisted product formation. (**a**) Schematic representation of the templating step at constant temperature. (**b**) Kinetics of tetramer formation at 45 °C with different starting concentrations of template ($\bar{c}_0$). Data includes concentrations of all complexes containing tetramers. (**c**) Templating observed over a broad temperature range. Large circles show data for reactions at $\bar{c}_0 = 120$ nM of template $\bar{0}_A\bar{0}_B\bar{0}_C\bar{0}_D$, small circles show the spontaneous formation ($\bar{c}_0 = 0$). The latter increases at T > 45 °C. Above 48 °C, binding of monomers to the template gets weaker, slowing down the rate of template assisted formation. This is consistent with the melting temperatures of the information domains (see *Figure 4—figure supplement 1*).

The online version of this article includes the following source data and figure supplement(s) for figure 4:

**Source data 1.** Source data for determination of thermal oscillation temperatures.
**Figure supplement 1.** Determination of thermal oscillation temperatures.
**Figure supplement 1—source data 1.** Source data for isothermal template assisted product formation.

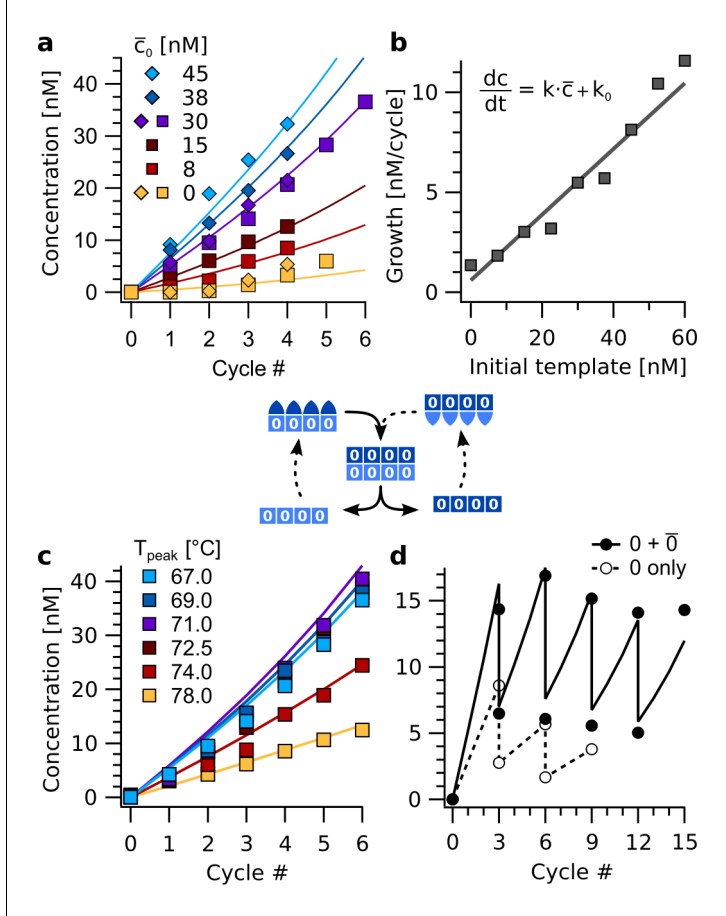

**Figure 5.** Exponential amplification of a restricted sequence subset with thermal oscillations. (**a**) Amplification time traces for concentration c for sequence 0000 during the first four to six cycles ($T_{peak}$ = 67 °C) for template ($\bar{0}_A\bar{0}_B\bar{0}_C\bar{0}_D$) concentrations $\bar{c}_0$ from 0 to 45 nM. The data was fitted using the cross-catalytic model from equation (1). Strands $0_A$, $\bar{0}_A$, $0_B$, ..., $\bar{0}_D$ were used at 200 nM concentration each. Data points show concentrations of complexes 4:4. (**b**) Initial reaction velocity as a function of initial template concentration $\bar{c}_0$. The data points show good agreement with the line calculated from the fits in panel **a**. (**c**) Amplification proceeded for peak temperatures below 74 °C. Above, backbone duplexes start to melt, and the complexes are no longer stable. The base temperature was 45 °C, reactions initially contained 30 nM of complex $\bar{0}_A\bar{0}_B\bar{0}_C\bar{0}_D$ as template. (**d**) Serial transfer experiment. The reaction containing strands $0_A$, $\bar{0}_A$, $0_B$, ..., $\bar{0}_D$ (black circles) survived successive dilution by a factor of 1/2 every three cycles at almost constant concentration. In contrast, a reaction with the same amount of template $\bar{0}_A\bar{0}_B\bar{0}_C\bar{0}_D$, but lacking monomers $\bar{0}_{A-D}$, fades out (open circles). The solid line shows the model from *Equation 1*.

The online version of this article includes the following source data for figure 5:

**Source data 1.** Source data for exponential amplification of a restricted sequence subset with thermal oscillations.

---

to be 0.43. The difference from the theoretical value of 0.50 was likely due to strands sticking to the reaction vessels before dilution. As a control, a reaction with the same initial concentration of template $\bar{0}_A\bar{0}_B\bar{0}_C\bar{0}_D$, but without monomers $\bar{0}_A$, $\bar{0}_B$, $\bar{0}_C$, $\bar{0}_D$, was subjected to the same protocol. As the control could not grow exponentially, it gradually died out (*Figure 5d*, open circles).

## Sequence replication

The above-mentioned reactions did amplify, but not replicate actual sequence information, as they only contained strands with $0/\bar{0}$ information domains. To study the replication of arbitrary sequences of binary code, replication reactions with all 16 strands encoding for '0' and '1' were performed. To discriminate sequences encoded in equally sized complexes and deduce error rates, we compared these results to those from different reaction runs with defects, that is lacking one or two of the

hairpin sequences required for the faithful replication of a particular template. Reference reactions contained all 16 strands ($0_A$, $\bar{0}_A$, $1_A$, $\bar{1}_A$, $0_B$, ..., $\bar{1}_D$) at 100 nM each, and were run for each of three different template sequences ($\bar{0}_A\bar{0}_B\bar{0}_C\bar{0}_D$, $\bar{0}_A\bar{1}_B\bar{0}_C\bar{1}_D$, and $\bar{0}_A\bar{0}_B\bar{1}_C\bar{1}_D$) (*Figure 6*). The product yields were quantified from reaction time traces, extracted by integrating the intensities of all gel bands containing tetramers with the labeled strand $0_A$.

Leaving out a single strand (reaction label "+++−", for example omitting $0_D$ for template $\bar{0}_A\bar{0}_B\bar{0}_C\bar{0}_D$) reduced the yield of full-size product to about 40 % (*Figure 6a, b*). The non-zero product yield with a missing strand is most likely due to the incorporation of the corresponding strand with an information domain mismatch (here $1_D$). This type of mismatch allows the hairpin backbone to form regardless, and the unfaithful product can propagate since both strands needed for an amplification of '1' at position D ($1_D$ and $\bar{1}_D$) are provided.

In particular during the first few cycles, mostly complex $0_A0_B0_C$:$\bar{0}_A\bar{0}_B\bar{0}_C\bar{0}_D$ (3:4) was detected in the gel, instead of the desired tetramer product (*Figure 6—figure supplement 1*). This was expected given the lack of strand $0_D$ and provides an upper limit on the error rate of the full replication. The fact that the full reaction produced almost no complexes 3:4 or 4:3 indicates that the incomplete product was indeed caused by the lack of a particular strand.

Removal of a further strand either directly next to the previous one ('++−−', missing strands $0_C$ and $0_D$) or not ('+−+−', missing strands $0_B$ and $0_D$) reduced the yield of product tetramers even further. Due to the periodic design those two variants represent all defective sets with two missing strands. Replication of the other two templates $\bar{0}_A\bar{1}_B\bar{0}_C\bar{1}_D$ and $\bar{0}_A\bar{0}_B\bar{1}_C\bar{1}_D$ produced very similar results. Product concentrations after six cycles are given in *Figure 6c* for each of the three templates as well as an average over the template sequences (horizontal lines). A single defect reduced the yield of tetramer complexes to about 40 %, two defects to 15–20 %, which is close to $0.4 \times 0.4 = 0.16 \simeq 15 - 20$ %, that is the combined probability of two independent mismatches.

## Replication fidelity

The observed rate of erroneous product formation can be attributed to the spontaneous background rate (*Figure 4b,c*, *Figure 5a,b* and *Figure 6b*). The reaction '+−+−' (dark green) amplified similarly to the untemplated reference reaction (solid line), as it did not contain any strands that could bind next to each other to the template and form a backbone duplex (*Figure 6b*). For the templated reactions '+++−' and '++−−', templating worked for partial sequences, producing intermediate yields.

The reduction in yield caused by a single defect (i.e. missing strand) to ~40 % (and to ~16 % for two defects) translates into a replication fidelity per information domain of ~60 %. The exact value for the replication fidelity is 62 % and can be calculated from *Figure 6b* by extracting the endpoint concentrations (blue vs. yellow line) and calculating $1 - \frac{14\,\text{nM}}{37\,\text{nM}} = 0.62$.

However, this is a worst-case estimation, and the replication fidelity is likely higher due to binding competition. The mutations caused by a single defect ('+++-') in *Figure 6b* were imposed by not providing strand $0_D$ for a template ending with $\bar{0}_D$ and only leaving the option to incorporate $1_D$ instead. For the full system ('++++'), however, with the presence of the matching strand, there is a binding competition for position D. Since the matching strand preferentially binds, the unfaithful incorporation of the wrong strand would be reduced. A similar effect of competition was observed in a protein-catalyzed ligation reaction (*Toyabe and Braun, 2019*). There, a comparable binding competition lead to a sevenfold decrease of the inferior ligation reaction in the presence of competition (*Figure 2a, b* therein). Therefore, we expect the real fidelity to be better than above lower bound estimate.

It is interesting to project and compare this per information domain replication fidelity to a per nucleotide replicator (i.e. polymerization). To do so, we define a threshold in the decrease of melting temperature per information domain as the criterion for when the replication mechanism is still functional. Then, we estimate how many point mutations in the information domain can maximally be tolerated to stay within this range of decrease in melting temperature. From this, we can calculate a hypothetical, corresponding per nucleotide fidelity to the measured information domain fidelity.

We compared the properties of the duplex $0$:$\bar{0}$ to duplexes $0$:$\bar{0}^*$, where $\bar{0}^*$ differs from $\bar{0}$ by $K$ point mutations. We assumed that within the temperature range of this replication mechanism (*Figure 7b*, gray box) a reduction in information domain melting temperature $T_m$ of the mutated

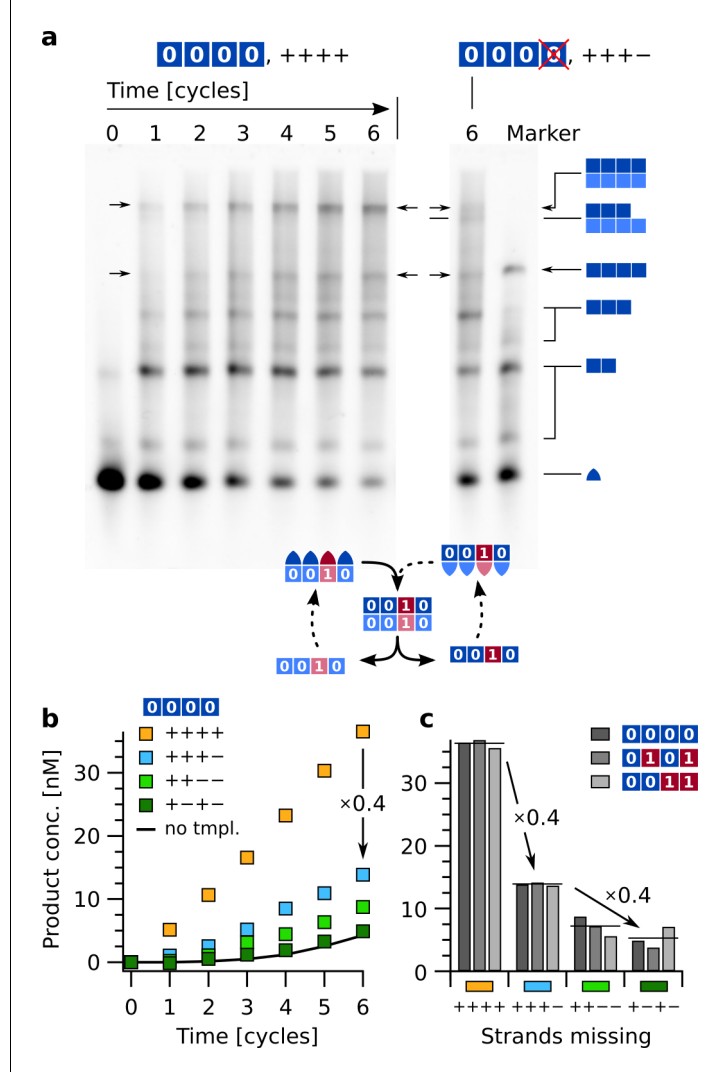

**Figure 6.** Sequence replication with thermal oscillations and fidelity check by forcing mutations from '0' to '1' at different locations. (a) Replication of sequence $0_A0_B0_C0_D$. Reactions were started with 15 nM initial template $\bar{0}_A\bar{0}_B\bar{0}_C\bar{0}_D$. All strands ($0_A$, $\bar{0}_A$, $1_A$, ..., $\bar{1}_D$) were present at 100 nM each. Native-PAGE results comparing the reaction of all 16 strands ('++++') with the reaction lacking strand $0_D$ ('+++−'). The defective set '+++−' mostly produced 3:4 complexes instead of 4:4 complexes (see schematics on the right). The overall yield of tetramer-containing complexes was greatly reduced. As size reference, the marker lane contained complexes $0_A0_B0_C0_D$, $0_A0_B0_C$, $0_A0_B$, and monomers $0_A$. The complete gel is presented in *Figure 6—figure supplement 1*. (b) Product concentration over time for the complete sequence network (yellow) and three defective sets with missing strands. Data was integrated by quantitative image analysis from electrophoresis gels using covalent markers on the $0_A$-strand counting all product complexes containing tetramers. Mutations of information in the product from '0' to '1' were induced by defective reactions that lacked strands $0_D$ ('+++−'), $0_C$ and $0_D$ ('++−−'), and $0_B$ and $0_D$ ('+−+−'). All reactions were initiated with 15 nM of $\bar{0}_A\bar{0}_B\bar{0}_C\bar{0}_D$. The solid line shows data from reaction '++++' without template. (c) End point comparison of reactions with templates $\bar{0}_A\bar{0}_B\bar{0}_C\bar{0}_D$ (panels a, b), $\bar{0}_A\bar{1}_B\bar{0}_C\bar{1}_D$, and $\bar{0}_A\bar{0}_B\bar{1}_C\bar{1}_D$ after six cycles. Horizontal lines indicate averages of the three template sequences. A single missing strand reduced product yield to about 40 %, two missing strands to 15–20 %.

The online version of this article includes the following source data and figure supplement(s) for figure 6:

**Source data 1.** Source data for sequence replication with thermal oscillations and fidelity check by forcing mutations from '0' to '1' at different locations.

**Figure supplement 1.** Extended electrophoresis gel image data.

**Figure supplement 1—source data 1.** Source data for extended electrophoresus gel image data.

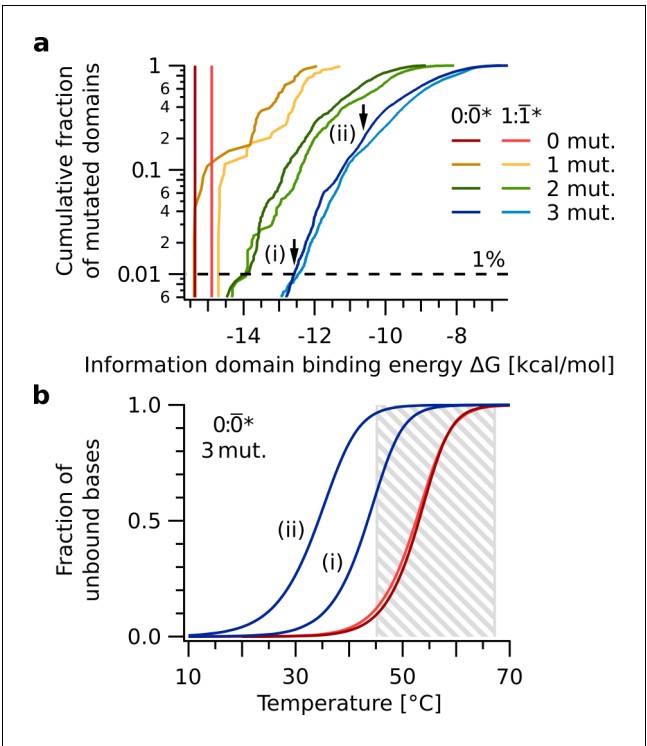

**Figure 7.** Sequence space analysis of information domain binding. The binding energies quantify the ability of the replication mechanism to discriminate nucleotide mutations. (a) Cumulative free energy distributions of information domain duplexes $0:\bar{0}$ (red), $1:\bar{1}$ (light red), as well as all $0:\bar{0}^*$ and $1:\bar{1}^*$ with up to three point mutations in $\bar{0}^*$ and $\bar{1}^*$ (yellow, green, blue). 99 % of duplexes $0:\bar{0}^*$ with three point mutations have free energies $\Delta G \geq$ -12.5 kcal/mol (dashed line), significantly weaker than that of $0:\bar{0}$ ($\Delta G$ = -15.4 kcal/mol). (b) Melting curves of information domain duplexes $0:\bar{0}$ (red), $1:\bar{1}$ (light red), and the two duplexes $0:\bar{0}^*$ indicated by arrows in panel a. Even the $0:\bar{0}^*$ duplex (i) at the low end of the $\Delta G$ distribution has a melting temperature of about 10 ˚C below that of $0:\bar{0}$. This difference in melting temperature destabilizes binding of the information domain and causes the replication mechanism to reject these sequences in the thermal oscillation regime between $T_{base}$ = 45 ˚C and $T_{peak}$ = 67 ˚C (gray box).

The online version of this article includes the following source data and figure supplement(s) for figure 7:

**Source data 1.** Source data for information domain binding energy statistics split into information domains containing terminal mutations and those with internal mutations only.

**Figure supplement 1.** Information domain binding energy statistics split into information domains containing terminal mutations and those with internal mutations only.

**Figure supplement 1—source data 1.** Source data for sequence space analysis of information domain binding.

duplex $0:\bar{0}^*$ by up to 10 ˚C compared to the original duplex $0:\bar{0}$ would be tolerated by the replication reaction. This was inferred from the width of the melting transition of duplex $0:\bar{0}$ (*Figure 7b*), where a shift of 10 ˚C corresponds to an increase of the unbound fraction from 0.08 at $T_{base}$ = 45 ˚C to 0.66 at 55 ˚C. In terms of free energies of the information domain duplex, this difference corresponds to $\Delta G(0:\bar{0}^*) \geq -12.5$ kcal/mol compared to $\Delta G(0:\bar{0}) = -15.4$ kcal/mol. 99 % of all duplexes $0:\bar{0}^*$, with $\bar{0}^*$ containing three point mutations, met that criterion (*Figure 7a*). Therefore, up to $K = 3$ point mutations can be allowed.

We will assume that the replication did not differentiate between information domain $\bar{0}$ and any information domain $\bar{0}^*$ if $\bar{0}$ and $\bar{0}^*$ differ by less than $K$ point mutations. The fidelity per information domain $p_K(N)$ is given by a cumulative binomial distribution:

$$p_K(N) = \sum_{k=0}^{K-1} \binom{N}{k} p^{N-k} (1-p)^k \tag{2}$$

Here, $N$ is the information domain length, and $p$ the per nucleotide replication fidelity. The reduction in binding energy of the information domain duplex $0:\bar{0}*$ and subsequent change in melting temperature was used as criterion to define the functionality of the replicator and to translate between a per information domain and a per nucleotide approach. As justified above, we calculate with $K=3$ mutations within the $N=15$ bases of the information domain, that is the replication can tolerate up to three mismatches in the information domain. From *Figure 6* we extracted a per information domain fidelity of $p_3(15)=0.62$, and deduce a per nucleotide fidelity of $p=85$ %. In fact, information domain duplexes $0:\bar{0}*$ with mutations at two internal bases all show similar properties as information domains with a total of three mutations (*Figure 7—figure supplement 1*). This refinement ($p_2(13)=0.62$) would increase the per nucleotide fidelity to $p=90$ %. We therefore estimate that a per nucleotide replication process would need a replication fidelity of 85–90 % to produce sequences with an error rate equivalent to the presented mechanism. Detailed calculations of the per nucleotide fidelities can be found in the supplementary information.

## Discussion

A cross-catalytic replicator can be made from short sequences and without covalent bonds under a simple non-equilibrium setting of periodic thermal oscillations. The replication is fast and proceeds within a few thermal oscillations of 20 min each. This velocity is comparable to other replicators (*Kindermann et al., 2005*), cross-ligating ribozymes (*Robertson and Joyce, 2014*), or autocatalytic DNA networks (*Yin et al., 2008*). The required thermal oscillations can be obtained by laminar convection in thermal gradients (*Braun et al., 2003*; *Salditt et al., 2020*), which also accumulates oligonucleotides (*Mast et al., 2013*). Depending on the envisioned environment, the mechanism could also be driven by thermochemical oscillations (*Ball and Brindley, 2014*) or convection in pH gradients (*Keil et al., 2017*). It should however be noted, that with the current state-of-the-art prebiotic chemistry regarding polymerization and ligation, the creation of >80 nt RNA is not yet understood.

It is likely that a slower prebiotic ligation chemistry could later fix the replication results over long timescales. Such an additional non-enzymatic ligation (*Stadlbauer et al., 2015*) that joins successive strands would relax the constraint that backbone duplexes must not melt during high-temperature steps. Early on, this is difficult to achieve in aqueous solution against the high concentration of water. In order to overcome this competition and to favor the reaction entropically by a leaving group, individual bases are typically activated by triphosphates (*Attwater et al., 2013*; *Horning and Joyce, 2016*) or imidazoles, which are especially interesting in this context since they can replicate RNA directly (*O'Flaherty et al., 2019*; *Zhou et al., 2019*). However, the required chemical conditions of enhanced $Mg^{2+}$ concentration hinder strand separation.

The overall replication fidelity is limited by the spontaneous bond formation rate between pairs of hairpin sequences, caused by the interaction of strands in free solution. At lower concentrations, as one would imagine in a prebiotic setting, this rate would decrease at the expense of an overall slower reaction. To some degree and despite ongoing design efforts, such a background rate is inherent to hairpin-fuelled DNA or RNA reactions (*Green et al., 2006*; *Krammer et al., 2012*; *Yin et al., 2008*).

The replication mechanism is expected to also work with shorter strands, as long as the order of the melting temperatures of the information domain and the backbone duplexes is preserved. Smaller strands would also be easier to produce by an upstream polymerization process, simply because they contain less nucleotides. In addition, binding of shorter information domain duplexes could discriminate even single base mismatches, resulting in an increased selectivity. It is not straightforward to estimate a minimal sequence length for the demonstrated mechanism. However, it is worth noting that it has been suggested that tRNA arose from two proto-tRNA sequences (*Hopfield, 1978*).

Pre-selection of nucleic acids for the presented hairpin-driven replication mechanism can be provided by highly sequence-specific gelation of DNA. This gel formation has been shown to be most efficient with double hairpin structures very similar to the tRNA-like sequences used in this study (*Morasch et al., 2016*). For our replication system, we have demonstrated this in *Figure 3* by showing the spontaneous formation of agglomerates and sedimentation under gravity if all molecules of the assembly are present. This self-selection shows a possible pathway how the system can emerge from random or semi-random sequences, for example in a flow or a convection system where the

molecules are selected as macroscopic agglomerate (*Mast et al., 2013*). Another selection pressure could stem from the biased hydrolysis of double-stranded nucleotide backbones, which favors assembled complexes over the initial hairpins (*Obermayer et al., 2011*).

The replication mechanism could serve as a mutable assembly strategy for larger functional RNAs (*Mutschler et al., 2015*; *Vaidya et al., 2012*). As an evolutionary route toward a more mRNA-like replication product with chemically ligated information domains, the mechanism would be supplemented by self-cleavage next to the information domains that cuts out the non-coding backbone duplexes, followed by ligation of the information domains. Both operations could potentially be performed by very small ribozymatic centers (*Dange et al., 1990*; *Szostak, 2012*; *Vlassov et al., 2005*).

The proposed replication mechanism of assemblies from tRNA-like sequences allows to speculate about a transition from an autonomous replication of successions of information domains to the translation of codon sequences encoded in modern mRNA (*Figure 1a*). Short peptide-RNA hybrids (*Griesser et al., 2017*; *Jauker et al., 2015*), combined with specific interactions between 3'-terminal amino acids and the anticodons, could have given rise to a primitive genetic code. The spatial arrangement of tRNA-like sequences that are replicated by the presented mechanism would translate into a spatial arrangement of the amino acid or short peptide tails that are attached to the strands in a codon-encoded manner (*Schimmel and Henderson, 1994*). The next stage would then be the detachment and linking of the tails to form longer peptides. Eventually, tRNA would transition to its modern role in protein translation. The mechanism thus proposes a hypothesis for the emergence of predecessors of tRNA, independent of protein translation. This is crucial for models of the evolution of translation, because it could justify the existence of tRNA before it was utilized in an early translation process. However, many questions around the evolutionary steps that created translation are still unclear.

Therefore, replication and translation could have, at an early stage, emerged along a common evolutionary trajectory. This supports the notion that predecessors of tRNA could have featured a rudimentary replication mechanism: starting with a double hairpin structure of tRNA-like sequences, the replication of a succession of informational domains would emerge. The interesting aspect is, that the replication is first encoded by hybridization and can later be fixed by a much slower ligation of the hairpins. The demonstrated mechanism could therefore jumpstart a non-enzymatic replication chemistry, which was most likely restricted in fidelity due to working on a nucleotide-by-nucleotide basis (*Robertson and Joyce, 2012*; *Szathmáry, 2006*).

# Materials and methods

**Key resources table**

| Reagent type (species) or resource | Designation | Source or reference | Identifiers | Additional information |
|---|---|---|---|---|
| Sequence-based reagent | $0_A$ | Biomers | | P - GCAGCGTTAATTCCCGC GCCTATCGGGAATGTAA CGCAGTGGGTAATAATG ACGATAGCCGTTCGGGA AAAGCGAACGGTATCG |
| Sequence-based reagent | $0_B$ | Biomers | | P - GCAGCGATACCGTTCG CTTTTCCCGAACGGCT ATCGCAGTGGGTAATA ATGAGCGAACTGTCGG TGCTTGCGACAGTGTCGC |
| Sequence-based reagent | $0_C$ | Biomers | | P - GCAGGCGACACTGTCG CAAGCACCGACAGTTC GCCAGTGGGTAATAAT GAGCGGTTCCTTGCGG AGTAGGCAAGGAATCCGC |
| Sequence-based reagent | $0_D$ | Biomers | | P - GCAGGCGGATTCCTTG CCTACTCCGCAAGGAA TCGCCAGTGGGTAATA ATGACGTTACATTCCC GATAGGCGCGGGAATTAACG |

*Continued on next page*

*Continued*

| Reagent type (species) or resource | Designation | Source or reference | Identifiers | Additional information |
|---|---|---|---|---|
| Sequence-based reagent | $\bar{0}_A$ | Biomers | | P - GCTGCGCATTAACGCG CTTGTCCCGCGTTAAT TGCGCTCATTATTACC CACTCGCTCTCGGCTG TTTTGCCCAGCCGAGCAGCG |
| Sequence-based reagent | $\bar{0}_B$ | Biomers | | P – GCTGCGTTGCATTGGC GATCAAAGCCAATGCG AACGCTCATTATTACC CACTCGCAATTAACGC GGGACAAGCGCGTTAATGCG |
| Sequence-based reagent | $\bar{0}_C$ | Biomers | | P - GCTGGTTGGAGAAGGC GAACAGCACGCCTTCC CAACCTCATTATTACCC ACTCGTTCGCATTGGC TTTGATC GCCAATGCAACG |
| Sequence-based reagent | $\bar{0}_D$ | Biomers | | P - GCTGCGCTGCTCGGCT GGGCAAAACAGCCGAG AGCGCTCATTATTACCC ACTGTTGGGAAGGCGT GCTGTTCGCCTTCTCCAAC |
| Sequence-based reagent | $1_A$ | Biomers | | P - GCAGCGTTAATTCCCG CGCCTATCGGGAATGT AACGCAAAAGAAGAGA AAGACGATAGCCGTTC GGGAAAAGCGAACGGTATCG |
| Sequence-based reagent | $1_B$ | Biomers | | P - GCAGCGATACCGTTCG CTTTTCCCGAACGGCT ATCGCAAAAGAAGAGA AAGAGCGAACTGTCGG TGCTTGCGACAGTGTCGC |
| Sequence-based reagent | $1_C$ | Biomers | | P - GCAGGCGACACTGTCG CAAGCACCGACAGTTC GCCAAAAGAAGAGAAA GAGCGGTTCCTTGCGG AGTAGGCAAGGAATCCGC |
| Sequence-based reagent | $1_D$ | Biomers | | P - GCAGGCGGATTCCTTG CCTACTCCGCAAGGAA TCGCCAAAAGAAGAGA AAGACGTTACATTCCC GATAGGCGCGGGAATTAACG |
| Sequence-based reagent | $\bar{1}_A$ | Biomers | | P - GCTGCGCATTAACGCG CTTGTCCCGCGTTAAT TGCGCTCTTTCTCTTC TTTTCGCTCTCGGCTG TTTTGCCCAGCCGAGCAGCG |
| Sequence-based reagent | $\bar{1}_B$ | Biomers | | P - GCTGCGTTGCATTGGC GATCAAAGCCAATGCG AACGCTCTTTCTCTTC TTTTCGCAATTAACGC GGGACAAGCGCGTTAATGCG |
| Sequence-based reagent | $\bar{1}_C$ | Biomers | | P - GCTGGTTGGAGAAGGC GAACAGCACGCCTTCC CAACCTCTTTCTCTTC TTTTCGTTCGCATTGG CTTTGATCGCCAATGCAACG |
| Sequence-based reagent | $\bar{1}_D$ | Biomers | | P- GCTGCGCTGCTCGGCT GGGCAAAACAGCCGAG AGCGCTCTTTCTCTTC TTTTGTTGGGAAGGCG TGCTGTTCGCCTTCTCCAAC |

*Continued on next page*

*Continued*

| Reagent type (species) or resource | Designation | Source or reference | Identifiers | Additional information |
|---|---|---|---|---|
| Sequence-based reagent | $0_A – Cy5$ | Biomers | | Cy5 -GCAGCGTTAATTCCCGC GCCTATCGGGAATGTAA CGCAGTGGGTAATAATG ACGATAGCCGTTCGGGA AAAGCGAACGGTATCG |
| Sequence-based reagent | $1_A – Cy5$ | Biomers | | Cy5 - GCAGCGTTAATTCCCG CGCCTATCGGGAATGT AACGCAAAAGAAGAGA AAGACGATAGCCGTTC GGGAAAAGCGAACGGTATCG |
| Sequence-based reagent | R (random) | Biomers | | NNNNNNNNNNNNNNNN NNNNNNNNNNNNNNNN NNNNNNNNNNNNNNNN NNNNNNNNNNNNNNNN NNNNNNNNNNNNNNNNNNNN |
| Sequence-based reagent | R (random) – Cy5 | Biomers | | Cy5 - NNNNNNNNNNNNNNNN NNNNNNNNNNNNNNNN NNNNNNNNNNNNNNNN NNNNNNNNNNNNNNNN NNNNNNNNNNNNNNNNNNNN |
| Software, algorithm | NUPACK | nupack.org | https://doi.org/ 10.1002/jcc.21596 | |
| Software, algorithm | ImageJ | ImageJ http://imagej. nih.gov/ij/ | RRID:SCR_002285 | |
| Software, algorithm | ImageJ stabilization plugin | http://www.cs. cmu.edu/~kangli/ code/Image_ Stabilizer.html | | |

## Strand design

DNA double-hairpin sequences were designed using the NUPACK software package (*Zadeh et al., 2011*). In addition to the secondary structures of the double-hairpins, the design algorithm was constrained by all target dimers. Candidate sequences were selected for optimal homogeneity of binding energies and melting temperatures. Backbone domains connecting consecutive strands (e.g. $0_A0_B0_C$) had to be the most stable bonds in the system, in particular more stable than between a template and a newly formed product complex (e.g. $0_B:\bar{0}_B$). On the other hand, hairpin melting temperatures had to be low enough to allow for a sufficient degree of thermal fluctuations. To reconcile this with the length of the strands, mismatches were introduced in the hairpin stems. The sequences of all strands are listed in *Supplementary file 1*.

## Thermal cycling assays

All reactions were performed in salt 20 mM Tris-HCl pH 8, 150 mM NaCl with added 20 mM MgCl$_2$. DNA oligonucleotides (Biomers, Germany) were used at 200 nM concentration per strand in reactions containing a fixed-sequence subset of eight strands (e.g. $0/\bar{0}$ only) and 100 nM per strand in reactions containing all 16 different strands.

Thermal cycling was done in a standard PCR cycler (Bio-Rad C1000). Reaction kinetics were obtained by running each reaction for different run times or numbers of cycles in parallel. The products were analyzed using native PAGE. The time between thermal cycling and PAGE analysis was minimized to exclude artifacts from storage on ice.

Template sequences were prepared using a two-step protocol. Annealing from 95℃ to 70℃ within 1 hr, followed by incubation at 70 ℃ for 30 min. Afterwards, samples were cooled to 2 ℃ and stored on ice. When assembling complexes containing paired information domains (*Figure 2*), samples were slowly cooled down from 70 to 25 ℃ within 90 min before being transferred onto ice. DNA double hairpins were quenched into monomolecular state by heating to 95 ℃ and subsequent fast transfer into ice water.

## Product analysis

DNA complexes were analyzed using native polyacrylamide gel electrophoresis (PAGE) in gels at 5 % acrylamide concentration and 29:1 acrylamide / bisacrylamide ratio (Bio-Rad, Germany). Gels were run at electric fields of 14 V/cm at room temperature. Strand $0_A/1_A$ was covalently labeled with Cy5. Cy5 fluorescence intensities were later used to compute strand concentrations. As an additional color channel, strands were stained using SYBR Green I dye (New England Biolabs). Complexes were identified by comparing the products obtained from annealing different strand subsets.

To correctly identify bands in the time-resolved measurements, gels were run with a marker lane. The marker contained strands $0_A$ (200 nM), $0_B$ (150 nM), $0_C$ (50 nM), and $0_D$ (100 nM), and was prepared using the two-step annealing protocol from 95 to 70 ˚C. The unequal strand concentrations ensured that the sample contained a mixture of mono-, di-, tri-, and tetramers.

Electrophoresis gels were imaged in a multi-channel imager (Bio-Rad ChemiDoc MP), image post processing, and data analysis were performed using a self-developed LabVIEW software. Post-processing corrected for inhomogeneous illumination by the LEDs, image rotation, and distortions of the gel lanes if applicable. Background fluorescence was determined from empty lanes on the gel, albeit generally low in the Cy5 channel.

For the determination of reaction yields, the intensities of all gel bands containing strands of the sequence length of interest were added up. For strings of four strands, these were the single tetramer as well as its complex with di- and tri- and tetramers. Single strands separated from their complements during electrophoresis (*Figure 2* and *Figure 6—figure supplement 1*).

## Thermal melting curves

Thermal melting curves were measured using either UV absorbance at 260 nm wavelength in a UV/Vis spectrometer (JASCO V-650, 1 cm optical path length), via quenching of the Cy5 label at the 5'-end of strand $0_A$ (excitation: 620–650 nm, detection: 675–690 nm), or using fluorescence of the intercalating dye SYBR Green I (excitation: 450–490 nm, detection: 510–530 nm). Fluorescence measurements were performed in a PCR cycler (Bio-Rad C1000). Samples measured via fluorescence were at 200 nM of each strand, those measured via UV absorption contained 1 µM total DNA concentration to improve the signal-to-noise ratio. Before analysis of the melting curves (*Mergny and Lacroix, 2003*), data were corrected for baseline signals from reference samples containing buffer and intercalating dye, if applicable.

## Self-assembly and sedimentation analysis

The samples were mixed in the replication buffer (150 mM NaCl, 20 mM MgCl$_2$, 20 mM Tris-HCl pH 8) at a total oligomer concentration of 5 µM, that is varying concentration per strand depending on the number of different strands in the configuration (4, 7, or 8). The microfluidic chamber was assembled with a custom cut, 500 µm thick, Teflon foil placed between two plane sapphires (*Figure 3—figure supplement 2*). Three Peltier elements (QuickCool QC-31–1.4-3.7AS, purchased from Conrad Electronics, Germany) were attached to the backside of the chamber to provide full temperature control. The chamber was initially flushed with 3M Novec7500 (3M, Germany) to avoid bubble formation. The samples were pipetted into the microfluidic chamber through the 0.5 mm channels using microloader pipette tips (Eppendorf, Germany). The chamber was then sealed with Parafilm and heated to 95 ˚C for 10 s to fully separate the strands and cooled rapidly (within 30 s) to 25 ˚C. Assembly and sedimentation were monitored for 20 hr on a fluorescence microscope (Axiotech Vario, Zeiss, Germany) with two LEDs (490 nm and 625 nm, Thorlabs, Germany) using a 2.5 x objective (Fluar, Zeiss, Germany). The observed sedimentation was independent of the attached dye and its position (*Figure 3—figure supplement 1c*). Prior to image analysis the image stacks were stabilized using an ImageJ plugin (*Li, 2008*). The ratio of sedimented fluorescence relative to the first frame after heating was used to quantify sedimentation (*Figure 3*). The sedimentation time-traces (*Figure 3b*) were fitted with a Sigmoid function to determine the final concentration increase $c/c_0$ (*Figure 3c*). The experiment was also performed with random 84 nt DNA strands at 5 µM total concentration to exclude unspecific agglomeration (*Figure 3—figure supplement 1c*).

## Acknowledgements

We gratefully acknowledge financial support the Deutsche Forschungsgemeinschaft (DFG) through the TRR 235 Emergence of Life (Project-ID 364653263) and the CRC 1032 NanoAgents (Project-ID 201269156). We thank for funding from the Graduate School 'Quantitative Bioscience Munich' (QBM). We appreciate the fruitful discussions in the Simons Collaboration on the Origins of Life, thank for the measurements by Thomas Rind and acknowledge discussions with Tim Liedl, Christof Mast and Lorenz Keil. We thank Filiz Civril, Adriana Serrão and Thomas Matreux for comments on the manuscript.

## Additional information

### Funding

| Funder | Grant reference number | Author |
| --- | --- | --- |
| Deutsche Forschungsgemeinschaft | TRR 235 | Alexandra Kühnlein Dieter Braun |
| Deutsche Forschungsgemeinschaft | Project-ID 364653263 | Alexandra Kühnlein Dieter Braun |
| Deutsche Forschungsgemeinschaft | CRC 1032 (A04) Project-ID 201269156 | Simon Alexander Lanzmich Dieter Braun |
| Deutsche Forschungsgemeinschaft | Student fellowship | Alexandra Kühnlein |
| Deutsche Forschungsgemeinschaft | Graduate school "Quantitative Bioscience Munich" | Alexandra Kühnlein |

The funders had no role in study design, data collection and interpretation, or the decision to submit the work for publication.

### Author contributions

Alexandra Kühnlein, Conceptualization, Data curation, Formal analysis, Visualization, Methodology, Writing - original draft, Writing - review and editing; Simon A Lanzmich, Conceptualization, Data curation, Software, Formal analysis, Visualization, Methodology, Writing - original draft; Dieter Braun, Conceptualization, Software, Funding acquisition, Methodology, Writing - original draft, Writing - review and editing

### Author ORCIDs

Alexandra Kühnlein (iD) https://orcid.org/0000-0001-9582-6304
Dieter Braun (iD) https://orcid.org/0000-0001-7751-1448

### Decision letter and Author response

Decision letter https://doi.org/10.7554/eLife.63431.sa1
Author response https://doi.org/10.7554/eLife.63431.sa2

## Additional files

### Supplementary files

- Source data 1. Nupack script used for the sequence design.

- Supplementary file 1. Sequences of all DNA strands used. Strand $0_A$ is 5'-labeled with Cy5, all other strands have a 5'-terminal phosphate. Solid underlines highlight hairpin loops, information domains are indicated by dashed underlines.

- Transparent reporting form

## Data availability

No data sets (e.g. sequencing data, clinical trial data etc.) were produced in this study. The source data files (Igor incl. macros) and data analysis (LabVIEW) tools used are provided as supporting fFiles (zip).

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

## Appendix 1

## Calculation of fidelity rate

Through the experiments shown in *Figure 6* we already know that the replication fidelity per information domain is 62 %. Now, we want to assume that the presented replication mechanism would translate into a base-by-base replication and look at (i) how tolerant would the replication be to point mutations at the information domain and (ii) given that threshold, how good would a base-by-base replication have to do to perform equally well, that is what per nucleotide fidelity would it need to have.

Question (i) is answered in *Figure 7*, where we see that on the 15 nt information domain we can allow up to three base mismatches to stay within the bounds of the temperature cycling (gray box, *Figure 7b*). In order to calculate how the measured replication fidelity per information domain translates into a hypothetical replication fidelity per nucleotide we assume a cumulative binomial distribution:

$$p_K(N) = \sum_{k=0}^{K-1} \binom{N}{k} p^{N-k} (1-p)^k$$

We know that the overall likelihood to get a 'correctly' replicated information domain is 62 %. From *Figure 7* we know that in a base-by-base replication, 'correctly' means with up to three mismatches. Therefore, we must find the number of combinatorial possibilities of spatially distributing 0, 1 or 2 mismatches on the 15 nt information domain (using $N = 15$ nucleotides and allowing up to $K = 3$ mismatches). Using this, we can determine the probability $p$ for a success, that is the correct replication of a single nucleotide, to meet the $p_K(N) = 0.62$ overall likelihood.

For $K = 3$ and $N = 15$, we measure the replication fidelity per information domain to be $p_{K(N)} = 0.62$. Therefore, we calculate:

$$
\begin{aligned}
\sum_{k=0}^{K-1} \binom{N}{k} p^{N-k}(1-p)^k = \sum_{k=0}^{2} \binom{15}{k} p^{15-k}(1-p)^k & = 0.62 \\
\binom{15}{0} p^{15}(1-p)^0 + \binom{15}{1} p^{14}(1-p)^1 + \binom{15}{2} p^{13}(1-p)^2 & = 0.62 \\
1p^{15} + 15p^{14}(1-p)^1 + 105p^{13}(1-p)^2 & = \\
= p^{15} + 15p^{14} - 15p^{15} + 105p^{13} - 210p^{14} + 105\,p^{15} & = \\
= 91p^{15} - 195p^{14} + 105p^{13} & = 0.62 \\
p = 0.853 & = 85\,\%
\end{aligned}
$$

From the information domain energy statistics shown in *Figure 7—figure supplement 1*, one can see that strands with two internal mutations behave nearly identical to strands with a total of three mutations (accepting internal and terminal mutations). Therefore, we simplify the calculation and only consider internal mutations.

Accordingly, we calculate for $K = 2$ and $N = \#all\,bases - \#terminal\,bases = 15 - 2 = 13$ and a per information domain fidelity $p_{K(N)} = 0.62$:

$$
\begin{aligned}
\sum_{k=0}^{K-1} \binom{N}{k} p^{N-k}(1-p)^k = \sum_{k=0}^{1} \binom{13}{k} p^{13-k}(1-p)^k & = 0.62 \\
\binom{13}{0} p^{13}(1-p)^0 + \binom{13}{1} p^{12}(1-p)^1 & = 0.62 \\
1p^{13} + 13p^{12}(1-p)^1 & = \\
= p^{13} + 13p^{12} - 13p^{13} & = \\
= -12p^{13} + 13\,p^{12} & = 0.62 \\
p = 0.900 & = 90\,\%
\end{aligned}
$$

Therefore, a comparable base-by-base replication would need a per nucleotide fidelity of 85–90 % to perform equally well as the presented replication mechanism.

