## [Decision Letter]

**Acceptance summary:**

We have found special interest in your new system, since it can serve as a new platform for building a replicator or an amplifier, and hence may help understanding how DNA/RNA self-replicate and pass on information.

**Decision letter after peer review:**

Thank you for submitting your article "tRNA sequences can assemble into a replicator" for consideration by *eLife*. Your article has been reviewed by three peer reviewers, one of whom is a member of our Board of Reviewing Editors, and the evaluation has been overseen by Patricia Wittkopp as the Senior Editor. The reviewers have opted to remain anonymous.

The reviewers have discussed the reviews with one another and the Reviewing Editor has drafted this decision to help you prepare a revised submission.

We would like to draw your attention to changes in our policy on revisions we have made in response to COVID-19 (https://elifesciences.org/articles/57162). Specifically, when editors judge that a submitted work as a whole belongs in *eLife* but that some conclusions require a modest amount of additional new data, as they do with your paper, we are asking that the manuscript be revised to either limit claims to those supported by data in hand, or to explicitly state that the relevant conclusions require additional supporting data.

Summary:

The authors describe a new replication system, driven by DNA molecules for which the sequence design was inspired by natural tRNAs. The new system assembles into four hairpins that can template the replication of complementary sequences. One of the main challenges in obtaining efficient (exponential) replication is originated from product inhibition, namely that the template-replicate duplexes are more stable than their single stranded forms, thus tend to have longer lifetimes and slow down the next replication cycles. In the current work, the authors developed a smart thermal strategy to drive the templates amplification. Their design offers a good mechanism for building a replicator or an amplifier and may serve as a simplified platform in understanding more about how DNA/RNA may self-replicate and pass on information. The experiments of templated kinetics and exponential replication seem to be conducted appropriately, and the article is well-written and supported on previous literature from the field.

The paper can be of interest to the growing community exploring the chemistry of the origin of life, and particularly replicative chemistry. The paper is recommended to be published after addressing the following comments.

Essential revisions:

1) For the Sequence replication, Figure 6b, what is the concentration of y-axis that is being reported? Is it the tetramer concentration (including 0A0B0C0D and 0¯A0¯B0¯C0¯D) or does it only include 0A0B0C0D? If it includes both, it should be specified, if not, can the authors explain why it is possible to form new 0A0B0C0D tetramers when there is no 0D fuels? Also, as all 4 reactions (++++, +++-, ++--, +-+-) starts with 15 nM of 0A0B0C0D tetramers, should they not begin with at least 15 nM in the 4 cases compared to the no template case?

2) Can the authors better explain the section of the calculation of Replication fidelity? As there are no mutations happening in the hairpins, wouldn't it make more sense for calculating the fidelity rate on information-domain-basis instead of on the nucleotide-basis? Also, the equation for fidelity per information domain 𝑝_𝐾_(𝑁) may need more explanation and clarification, it would be helpful, thus, to add a section in the supplementary information clearing up the definition of fidelity rate (of the final tetramers, how many of them are 0A0B0C0D) and how these equations are derived. The 71 % fidelity per information domain caused by the 40 % decrease is also a bit confusing, as in this case, all these products should be errors, doesn't it translate to 60 % of leak for such cases?

3) No comments are made along the paper regarding the possibility of non-nucleic acid molecules to replicate, but many recent studies have highlighted this possibility. As a minimum, to complete the discussion it would be nice to refer the reader to review papers on this topic. In addition, since the replication process described in this paper is based on self-assembly, some references to previous works where self-assembly and replication were also intimately related are missing.

4) By reading the end of the Abstract (and from the title) one may understand that today's tRNA sequences, or very similar sequences, could confer any advantages with respect to replication and selection. As this point is not directly demonstrated in the paper, the authors could consider a more conservative discussion of this issue and clarify how far this argument holds.

5) The statement "Here, we implemented the system with DNA for practical reasons. Nevertheless, due to short heating times and moderate magnesium concentrations, we also estimate that an RNA version can survive for weeks (Li and Breaker, 1999)" may be true according to that reference, but it would have so important implications that it requires some experimental verification. Otherwise, it would be better to be more conservative about this point.

---

## [Author Response]

Essential revisions:1) For the Sequence replication, Figure 6b, what is the concentration of y-axis that is being reported? Is it the tetramer concentration (including 0A0B0C0D and 0¯A0¯B0¯C0¯D) or does it only include 0A0B0C0D? If it includes both, it should be specified, if not, can the authors explain why it is possible to form new 0A0B0C0D tetramers when there is no 0D fuels? Also, as all 4 reactions (++++, +++-, ++--, +-+-) starts with 15 nM of 0A0B0C0D tetramers, should they not begin with at least 15 nM in the 4 cases compared to the no template case?

In these settings, as the reviewers noted correctly, Figure 6b shows indeed the product concentration. As the label sits on strand 0A, the template is unlabeled and only the formation of product (containing 0A) is recorded. This also becomes clear in Figure 6a, where the labeled strand 0A at timepoint zero shows no product complexes but over time gets incorporated in the replicated product complexes. Therefore, the starting concentration of the product is indeed 0 nM, as shown in the Figure 6b. Only the unlabeled template 0¯A0¯B0¯C0¯D is present in the first cycle at a concentration of 15 nM.

To study the replication of information, the performed reactions contained all 16 fuel strands (i.e. 0A,0B,0C,0D, 0¯A,0¯B,0¯C,0¯D, 1A,1B,1C,1D, 1¯A,1¯B,1¯C,1¯D), and started with one of the three templates 0¯A0¯B0¯C0¯D, 0¯A0¯B1¯C1¯D, 0¯A1¯B0¯C1¯D. We then compare the amount of product for the complete reaction (“++++”) to the output for reactions lacking one or more of the fuel strands required to replicate the template after six oscillations. For example, in the case “+++-“, when providing the template 0¯A0¯B0¯C0¯D, we omitted 0D and thus forced the reaction to create a mutation from “0” to “1” at position D. For the case “++--“ and the template 0¯A0¯B0¯C0¯D, both 0C and 0D were omitted, which forced two mutations from “0” to “1” at positions C and D.

One might also ask why we did not use differentially labeled strands. We deliberately decided against this, as we wanted to keep the labeled strand constant and not introduce additional labels to avoid differences in binding due to potential stacking interactions of or with the dyes.

To better show all this, we have changed the y-label of Figure 6b and adapted the figure legend.

In all cases, when a fuel strand required for the correct product formation is removed, e.g. removing 0D when providing template 0¯A0¯B0¯C0¯D, the product yield of tetramers is reduced to about 40 %. Since it is impossible to form the correct product, the detected tetramers must contain mismatches at the position where the correct strand is missing. In principle, this can be two kinds of mismatches:

i) The incorporation of the correct information domain but at the wrong position, leading to an insufficiently formed backbone. For example, 0B could be incorporated at the position of 0D. But location mismatches of this type will break up during the next temperature cycle as 0B could only bind at the information domain but is incompatible along the hairpins and would therefore not lead to the formation of a tetramer. Such a reaction would not be reflected in the final yield of tetramers.

ii) The second type of mismatch will in contrast alter the final concentration of tetramers. A correctly formed backbone is created, but an information domain mismatch occurs, e.g. 1D is incorporated at the position of 0D. This tetramer with the mutation from “0” to “1” at position D will not break up during the temperature cycles due to its correctly formed backbone. Therefore, we enter the next round of replication with an additional tetramer with a correctly formed backbone but a different sequence, e.g. here 0A0B0C1D. This will continue to replicate since for this tetramer all fuel strands are available. From now on, 0A0B0C1D can act as a template for an unfaithful replication. We argue that mismatches of type (ii) are most likely the reason for a tetramer product yield of 40 % despite one missing strand.

In Figure 6c, we also tested above scheme for two other templates 0¯A1¯B0¯C1¯D and 0¯A0¯B1¯C1¯D which behaved in a very similar manner and showed that the probability of two mutations is in good approximation with the squared probability of a single mutation (0.4×0.4 = 0.16 ≃ 15 − 20 %). This indicates that the processes causing the non-zero product yields in first approximation are independent, which matches with our explanation of mismatch type (ii). Due to the periodicity of the design, the two defective sets (“++--” and “+-+-“) cover the whole combinatorial space of two mismatches.

Clarifying those points, we now write:

“Reference reactions contained all 16 strands (0A, 0¯A, 1A, 1¯A, 0B, …, 1¯D) at 100 nM each, and were run for each of three different template sequences (0¯A0¯B0¯C0¯D, 0¯A1¯B0¯C1¯D, and 0¯A0¯B1¯C1¯D) (Figure 6). […] A single defect reduced the yield of tetramer complexes to about 40 %, two defects to 15–20 %, which is close to 0.4×0.4=0.16≃15−20 %, i.e. the combined probability of two independent mismatches.”

2) Can the authors better explain the section of the calculation of Replication fidelity? As there are no mutations happening in the hairpins, wouldn't it make more sense for calculating the fidelity rate on information-domain-basis instead of on the nucleotide-basis? Also, the equation for fidelity per information domain pK(N) may need more explanation and clarification, it would be helpful, thus, to add a section in the SI clearing up the definition of fidelity rate (of the final tetramers, how many of them are 0A0B0C0D) and how these equations are derived. The 71 % fidelity per information domain caused by the 40 % decrease is also a bit confusing, as in this case, all these products should be errors, doesn't it translate to 60 % of leak for such cases?

We indeed agree that the replication fidelity was estimated wrongly by assigning the 100 % to the sum of the concentrations in the calculation of the fraction of perfect matches. We therefore revised the numbers given regarding the fidelity of the replicator.

This reduces our estimation of the replicator’s fidelity from 71 % down to 62 %. However, we want to point out that in the way we test for mutations, competition for binding sites is neglected, which is why a replication fidelity of 62 % is a lower bound estimation.

As correctly noted by the reviewers, the replication fidelity is defined by how much of the replicated information is replicated accurately. To stay with the example from (M1) and Figure 6b, this means how many of the product tetramers from template 0¯A0¯B0¯C0¯D and “++++” do actually contain the accurate product sequence 0A0B0C0D. In the experiments shown in Figure 6b, c we have determined the probability of mutations in the absence of 0D where we obtain a ~40 % yield. The exact value for the replication fidelity of 62 % can directly be calculated from Figure 6b by extracting the endpoint concentrations (blue line vs. yellow lane) and calculating 1− 14 nM37 nM = 0.62. Please note that for the calculation of the replication fidelity we now use a 2-digit precision, whereas for simplicity we stick with a 1-digit precision in Figure 6.

The experiments presented in Figure 6 measured the rate of incorporation of 1D if no 0D was present and there was no competition in binding. For the case of provided template 0¯A0¯B0¯C0¯D – in Figure 6c different templates are analyzed – this means that the faithfully replicated products without competition amount to a ratio of 62 %, a little bit lower than initially stated in the manuscript where we incorrectly put the 100 % reference to the sum of both above concentrations.

We think this is a worst-case scenario. The mutations in the “+++-” case of Figure 6b were forced from a template ending with 0¯D and could only bind 1D as the optimal fuel 0D was not provided. In the full system, the presence of the matching fuel strand, which binds preferentially at position D would have reduced the unfaithful incorporation of the wrong strand. We have seen a similar effect of competition for a protein-catalyzed ligation reaction (Toyabe and Braun, 2019). There, a comparable binding competition lead to a 7-fold decrease of the inferior ligation reaction in the presence of competition (Figure 2a, b therein).

Following this argumentation, we expect that the mutations from “0” to “1” would occur much less under competition, when the fuel for “0” is provided in the mutation experiment shown in Figure 6b. How much this will actually be the case is however hard to estimate as the analysis cannot distinguish between sequences.

The calculation of the replication fidelity per nucleotide is a projection. The aim in calculating a number for the per nucleotide replication fidelity is to compare our work to other studies, which are in comparison base-by-base replicators and provide a number for the replication fidelity per nucleotide.

Through the experiments shown in Figure 6 we already know that the replication fidelity per information domain is 62 %. Now, we assume that the presented replication mechanism would translate into a base-by-base replication and look at (i) how tolerant would the replication scheme be to point mutations at the information domain and (ii) given that threshold, how well would a base-by-base replication have to perform in terms of per nucleotide fidelity.

We explain the reasoning in calculating the per nucleotide fidelity more thoroughly now in the manuscript. We also added a section, where we provide more detail on why a binomial distribution is used and how the exact numbers for the replication fidelity per nucleotide are calculated.

“Question (i) is answered in Figure 7, where we see that on the 15 nt information domain we can allow up to three base mismatches to stay within the bounds of the temperature cycling (gray box, Figure 7b). We make this clearer in the manuscript now. In order to calculate how the measured replication fidelity per information domain translates into a hypothetical replication fidelity per nucleotide we assume a cumulative binomial distribution:

pK(N) = ∑k=0K−1(Nk) pN−k(1−p)k We know that the overall likelihood to get a “correctly” replicated information domain is 62 %. From Figure 7 we know that in a base-by-base replication, “correctly” translates to three mismatches to sustain the replication. Therefore, we must find the number of combinatorial possibilities of spatially distributing 0, 1 or 2 mismatches on the 15 nt information domain (using N=15 nucleotides and allowing up to K=3 mismatches). Using this, we can determine the probability p for a success, i.e. the correct replication of a single nucleotide, to meet the pK(N)=0.62 overall likelihood.”

For N=15 and K=3, the cumulative binomial distribution p3(15)=0.62 can be solved for p (the per-nucleotide fidelity needed) , which yields p=0.85. When neglecting the terminal mismatches (Figure 7—figure supplement 1), we calculate p=0.90 after solving p2(13)=0.62. Therefore, a comparable base-by-base replication would need a per nucleotide fidelity of 85-90 % to perform equally well as the presented replication mechanism.

Reflecting the above discussion, we now write in the manuscript:

“The reduction in yield caused by a single defect (i.e. missing strand) to ~40 % (and to ~16 % for two defects) translates into a replication fidelity per information domain of ~60 %. […] Detailed calculations of the per nucleotide fidelities can be found in the subsection “Calculation of fidelity rate”.”

In addition, we included the following extra section:

**“**Calculation of fidelity rate

Through the experiments shown in Figure 6 we already know that the replication fidelity per information domain is 62 %. […] Therefore, a comparable base-by-base replication would need a per nucleotide fidelity of 85 - 90 % to perform equally well as the presented replication mechanism.”

We also adjusted the number given for the replication fidelity per nucleotide in the Abstract to 88 % and reformulated the sentence to minimize confusion about the per information domain and per nucleotide fidelity. We now write:

“The molecular assembly could encode and replicate binary sequence information with a replication fidelity corresponding to 85 - 90 % per nucleotide.”

3) No comments are made along the paper regarding the possibility of non-nucleic acid molecules to replicate, but many recent studies have highlighted this possibility. As a minimum, to complete the discussion it would be nice to refer the reader to review papers on this topic. In addition, since the replication process described in this paper is based on self-assembly, some references to previous works where self-assembly and replication were also intimately related are missing.

This is indeed a very good point – and apologies for our nucleotide-centered point of view. We now add to the manuscript:

“Apart from nucleotide-based replicators, very interesting replication systems using non-covalent interactions have been developed with non-biological compounds (Bottero et al., 2016; Sadownik & Philp, 2008, Tjivikua et al., 1990,), peptide-based approaches (Altay et al., 2017; Bourbo et al., 2011; Carnall et al., 2010; Lee et al., 1996; Rubinov et al., 2012) and peptide nucleic acids (Ura et al., 2009). We also want to point to several instructive reviews about the state-of-the-art systems chemistry regarding self-replication (Adamski et al., 2020, Ashkenasy et al., 2017, Kosikova and Philp, 2017).”

4) By reading the end of the Abstract (and from the title) one may understand that today's tRNA sequences, or very similar sequences, could confer any advantages with respect to replication and selection. As this point is not directly demonstrated in the paper, the authors could consider a more conservative discussion of this issue and clarify how far this argument holds.

We understand the reviewers’ reservations about this point. We want to stress that we do not claim that today’s tRNA sequences have any advantages in today’s replication or selection. We merely argue that our experiments support a hypothesis under which tRNA which is one of the most ancient molecules of modern biology might have transformed its role over time. While it today is responsible for the translation of proteins, it might much earlier, in a maybe slightly different form, have been involved in a molecular replication scheme, like the one presented in this manuscript.

We have therefore reformulated the sentence in the Abstract to make it clear that our argument is about a connection in very early replication mechanisms. Of course, it is difficult to argue that modern tRNA sequences are very close to ancient ones. But we hope that our discussion of the replication mechanism on the basis of melting temperatures and the kinetics of hybridization makes it clear that we do not rely on a very specific sequence, but merely on hybridization and a conserved order of melting temperatures.

We now write in the Abstract:

“The replication by a self-assembly of tRNA-like sequences suggests that early forms of tRNA could have been involved in molecular replication. This would link the evolution of translation to a mechanism of molecular replication.”

5) The statement "Here, we implemented the system with DNA for practical reasons. Nevertheless, due to short heating times and moderate magnesium concentrations, we also estimate that an RNA version can survive for weeks (Li and Breaker, 1999)" may be true according to that reference, but it would have so important implications that it requires some experimental verification. Otherwise, it would be better to be more conservative about this point.

Krammer et al. reported a much more primitive replicator with single and not double hairpins and therefore only half the sequence length (Krammer et al., 2012). It is important to note that in this 2012 study the replicator was implemented in RNA. After remodeling the replicator in DNA, we can say that for both replicators, the RNA and the DNA version, we could explain their behavior based on hybridization, and did not have to include any extra considerations for the RNA version.

We also included an overview (see Figure 1—figure supplement 1b) over the predicted secondary structures and the free energies for an RNA version of the presented replicator, when substituting every 'T' with a 'U', using NUPACK (Zadeh et al., 2011). The secondary structure is identical to the DNA version (compare with Figure 1—figure supplement 1a). Only the free energies are slightly higher (+30 %) which could be compensated by a reduction of salt concentrations for an RNA implementation.

Therefore, we argue that the replication scheme can readily be implemented in RNA. Even though the timescales on which Krammer et al. performed their experiments are much shorter, the initial heating step to 95 °C (20 mins > 80 °C) is identical and would arguably have the strongest effect on RNA stability compared to the moderate temperatures during cycling, in both Krammer et al. (10 °C (27 s) - 40 °C (3 s)) and this study (45 °C (20mins) - 67 °C (20 s)).We also want to quote another recent study looking at the hydrolysis of RNA by Mariani et al. They determined the half-life for a 10 nt RNA at 10mM Mg^2+^ at 90 °C to be seven days. For a 30 nt RNA under the same conditions they measure 16 % unspecific degradation after seven days (Mariani et al., 2018). Even though our Mg^2+^ concentration is 2-fold higher, we are operating at much more moderate temperatures. We also want to mention a recent study from our own lab, where replication with a 200 nt ribozyme was performed at much higher Mg^2+^ concentration (50 mM) including temperature spikes (Salditt et al., 2020), which were tolerated well and confirm the hydrolysis studies cited in this manuscript.

We understand that RNA stability at high magnesium concentration and high temperature is critical, but as the time of exposure at high temperature in the presented replication scheme is limited, we would stick with our claim however in a reformulated form. Although, we now elaborate more on the previous implementation with RNA. We now write:

"Here, we implemented the system with DNA and not RNA as done previously (Krammer et al., 2012). Both, in the design and the implementation we did not see significant differences between the two versions. Because of the simpler and more inexpensive synthesis of the 82-84 nt long sequences we now implemented the replicator in DNA. Due to short heating times and moderate magnesium concentrations, we estimate that an RNA version could survive for days if not weeks (Li & Breaker, 1999, Mariani et al. 2018). The most critical step regarding the RNA stability would be the initial temperature spike to 95 °C, which remains unchanged from our previous study (Krammer et al., 2012) and did not prove critical. In Figure S1 we also show that an RNA version behaves structurally identical to the implemented DNA version."